# Growth phase-dependent reorganization of cryptophyte photosystem I antennae
Shumeng Zhang [1,3], Long Si[1,2,3], Xiaodong Su[1], Xuelin Zhao[1], Xiaomin An[1] & Mei Li [1] ✉

Photosynthetic cryptophytes are eukaryotic algae that utilize membrane-embedded chlorophyll a/c binding proteins (CACs) and lumen-localized phycobiliproteins (PBPs) as their light-harvesting antennae. Cryptophytes go through logarithmic and stationary growth phases, and may adjust their light-harvesting capability according to their particular growth state. How cryptophytes change the type/arrangement of the photosynthetic antenna proteins to regulate their light-harvesting remains unknown. Here we solve four structures of cryptophyte photosystem I (PSI) bound with CACs that show the rearrangement of CACs at different growth phases. We identify a cryptophyte-unique protein, PsaQ, which harbors two chlorophyll molecules. PsaQ specifically binds to the lumenal region of PSI during logarithmic growth phase and may assist the association of PBPs with photosystems and energy transfer from PBPs to photosystems.

Oxyphototrophic organisms sustain most of the life forms on earth by absorbing solar energy and converting it into the chemical energy and releasing oxygen. Photosystems I and II (PSI and PSII) are two thylakoid membrane-embedded complexes essential for the photosynthetic light reactions. Both photosystems contain a structurally conserved core complex and a highly variable antenna system. The antenna proteins increase the light harvesting capability of the photosystem core where the charge separation occurs. While most eukaryotic phototrophs utilize the membrane-embedded light-harvesting complexes (LHCs) as peripheral antennae for both photosystems, prokaryotic cyanobacteria contain huge membrane-extrinsic phycobilisomes (PBSs) serving as the antenna system of PSII[1–3]. Red algae possess both PBSs and LHCs as antennae for PSII and PSI, respectively[4,5]. The variable antenna systems allow the efficient photosynthesis of different organisms, and are essential for them to adapt to their respective habitats[6].

Photosynthetic cryptophytes are unicellular organisms that acquired their chloroplasts from red algae by secondary endosymbiosis millions years ago[7]. These microalgae utilize both membrane-embedded LHCs and membrane-extrinsic phycobiliproteins (PBPs) as antennae[8]. However, PBPs in cryptophytes form rhombic α1α2ββ-tetramers[9,10] and are located at the thylakoid lumen[11], differing from the stroma-localized disk-shaped PBPs in cyanobacteria and red algae, which are usually composed of three αβ-heterodimers[1]. Cryptophyte LHCs are similar to those in red algae and diatom, but contain unique carotenoid alloxanthin, as well as chlorophyll (Chl) a and c2, thus were named ACP (alloxanthin and chlorophyll a/c

binding protein)[12] or more generally termed CAC (chlorophyll a/c binding protein)[13].

Cryptophyte algae are single-cellular organisms that go through logarithmic and stationary growth phase (L-phase and S-phase)[14,15]. While L-phase cells contain higher amounts of PBPs and exhibit high photosynthetic efficiency[14,15], cells in S-phase are characterized by greatly reduced amount of PBPs[14]. The cryptophyte cells may regulate their light-harvesting capability via utilizing a specialized antenna system according to their particular growth phase. The recently determined structures of PSI-ACPI from *Chroomonas placoidea* (*Cp*PSI-ACPI) provide a basis for understanding the assembly and pigment arrangement of CACs in cryptophyte PSI at L-phase[12]. However, the precise details of how the antennae of PSI are organized at S-phase remain unknown.

Here, we solved four structures of cryptophyte PSI-CAC complexes purified from *Rhodomonas salina* (*Rs*) cells grown at L- and S-phase. Comparison of the four structures reveals how cryptophytes adjust their PSI antennae in response to the different growth state. Our results provide structural basis for detailed understanding of the regulatory mechanism of photosynthetic organisms to cope with the environmental stresses.

## Results
### Overall structure of PSI-CACs

We cultured cryptophyte *Rhodomonas salina* (*Rs*) cells in F/2 medium[16] and measured their growth curve by continuously counting the cell number for 13 days. Our result showed that *R. salina* cells first grow exponentially and

[1]Key Laboratory of Biomacromolecules (CAS), National Laboratory of Biomacromolecules, CAS Center for Excellence in Biomacromolecules, Institute of Biophysics, Chinese Academy of Sciences, Beijing, China. [2]University of Chinese Academy of Sciences, Beijing, China. [3]These authors contributed equally: Shumeng Zhang, Long Si. ✉e-mail: meili@ibp.ac.cn

then enter the S-phase at the 9th day (Supplementary Fig. 1a). Moreover, we found that *R. salina* cells at S-phase showed reduced absorption at 545 nm (Supplementary Fig. 1b), indicating that the amount of PE545 (PBP in *R. salina*) decreases in S-phase cells. These observations are in agreement with previously reported results[14,17,18]. The *R. salina* cells grown for six and 12 days (corresponding to L-phase and S-phase) were harvested, and utilized to isolate PSI-CAC complexes. We then solved four structures of PSI-CAC complexes purified from cells in both growth phases, containing either 14 or 11 CACs, and termed them as PSI-14CAC$_{L-phase}$ and PSI-11CAC$_{L-phase}$ for L-phase models, and PSI-14CAC$_{S-phase}$ and PSI-11CAC$_{S-phase}$ for S-phase models (Figs. 1–2, Table 1, Supplementary Figs. 1–6).

Two PSI-14CAC structures closely resemble each other, containing 14 CAC subunits which encircle the core complex. The 14 CACs were named CAC-a to CAC-n in a clockwise direction when looked at the lumenal side (Figs. 1–2). Eleven CACs (CAC-a to CAC-k) constitute the inner antenna layer surrounding the core as a closed ring, whereas CAC–l/m/n form the outer layer covering CAC-a/b/c. One subunit (termed chain-s) contains one transmembrane helix and binds three Chl a, two Chl c2 and three α-carotene molecules (Supplementary Figs. 6a, 7a). Chain-s is sandwiched by the inner CAC-a/b/c and the outer CAC-l/m/n (Fig. 1), and forms multiple hydrogen bond interactions with CAC-a/b/l/m and PsaF (Supplementary Fig. 7b, c). The only difference between the two PSI-14CAC structures is that in the PSI-14CAC$_{L-phase}$ complex, one four-helix-bundle protein situates at the lumenal side, while in the PSI-14CAC$_{S-phase}$ structure, it appears to be missing (Figs. 1–2). Based on our high-quality cryo-EM map and transcriptome sequencing results (Supplementary Fig. 6b, c), we determined the identity of the lumen-localized protein, which is an uncharacterized yet highly conserved protein in cryptophytes (Supplementary Fig. 8). The overall folding of this protein is similar to the PSII extrinsic subunit PsbQ

(Supplementary Fig. 9). In addition, with the exception of PsaQ, names from PsaA to PsaS have been previously assigned to PSI subunits[19–21]; we therefore termed it PsaQ in alphabetical order.

Except CAC-a and CAC–h, all CACs are grouped as four hetero-trimers. Consistent with the trimeric organization of CACs, both PSI-11CAC structures lack one CAC trimer, but at distinct positions (Fig. 2). PSI-11CAC$_{L-phase}$ loses the inner trimer-e/f/g which attaches to PsaO, whereas PSI-11CAC$_{S-phase}$ lacks the outer trimer-l/m/n, which links to the inner layer through chain-s. Accordingly, PsaO is untraceable in the PSI-11CAC$_{L-phase}$ structure, and chain-s is absent in our PSI-11CAC$_{S-phase}$ structure. While PsaQ is also absent in our PSI-11CAC$_{S-phase}$ structure, it binds to PSI-11CAC$_{L-phase}$ at the same lumenal position as that in PSI-14CAC$_{L-phase}$ (Fig. 2).

## Structural features of *R. salina* CACs

We identified all fourteen CAC proteins in our structures, based on our cryo-EM density and transcriptome sequencing analysis (Supplementary Fig. 10). All CACs adopt the same overall folding, containing three transmembrane helices termed helix B, C and A from the N- to C-terminus. It is noteworthy that CAC-h constitutes a RedCap (red lineage chlorophyll a/b-binding-like protein)[22], and exhibits slightly different conformation and binds fewer pigment molecules compared to canonical CACs which possess 11-15 chlorophylls and five carotenoids (Supplementary Fig. 11, Supplementary Table 1). CAC-a is located slightly distant from CAC-b, as the N-terminal loop of chain-s separates the two CACs at the stromal side (Fig. 1). The remaining 12 CACs form four well-superposed trimers. CACs at equivalent positions in four trimers display high similarities in both protein folding and pigment arrangement, whereas CACs at different positions within the same trimer showing their respective characteristics (Fig. 3a, b, Supplementary Fig. 10). Specifically, the first CACs (CAC-b/e/i/l)

**Fig. 1 | Overall structure of the PSI-14CAC$_{L-phase}$ complex.** Density map of *Rs*PSI-14CAC$_{L-phase}$ complex viewed from different sides. For clarity, CACs are labeled as one letter with "CAC" being omitted.

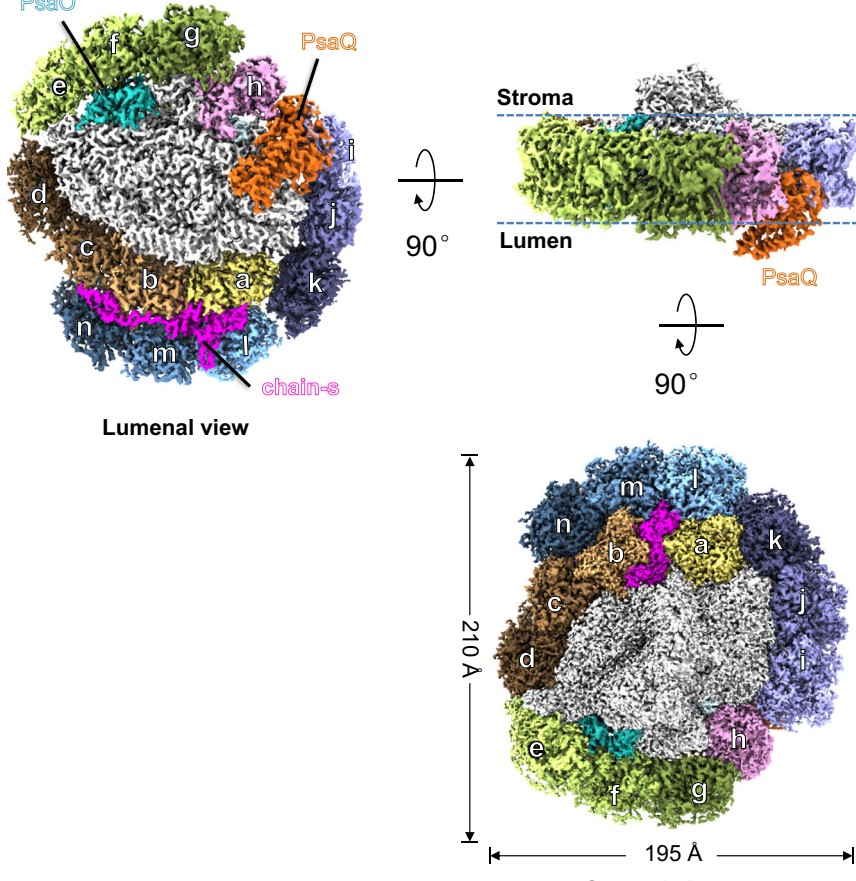

**Fig. 2 | Structures of four types of *Rs*PSI-CAC complexes.** Structures are viewed from the lumenal side and the membrane plane, respectively. Four CAC trimers as well as PsaO and chain-s are labeled in PSI-14CAC_{S-phase} structure, whereas PsaQ is indicated in the PSI-14CAC_{L-phase} structure. The color code is the same as in Fig. 1.

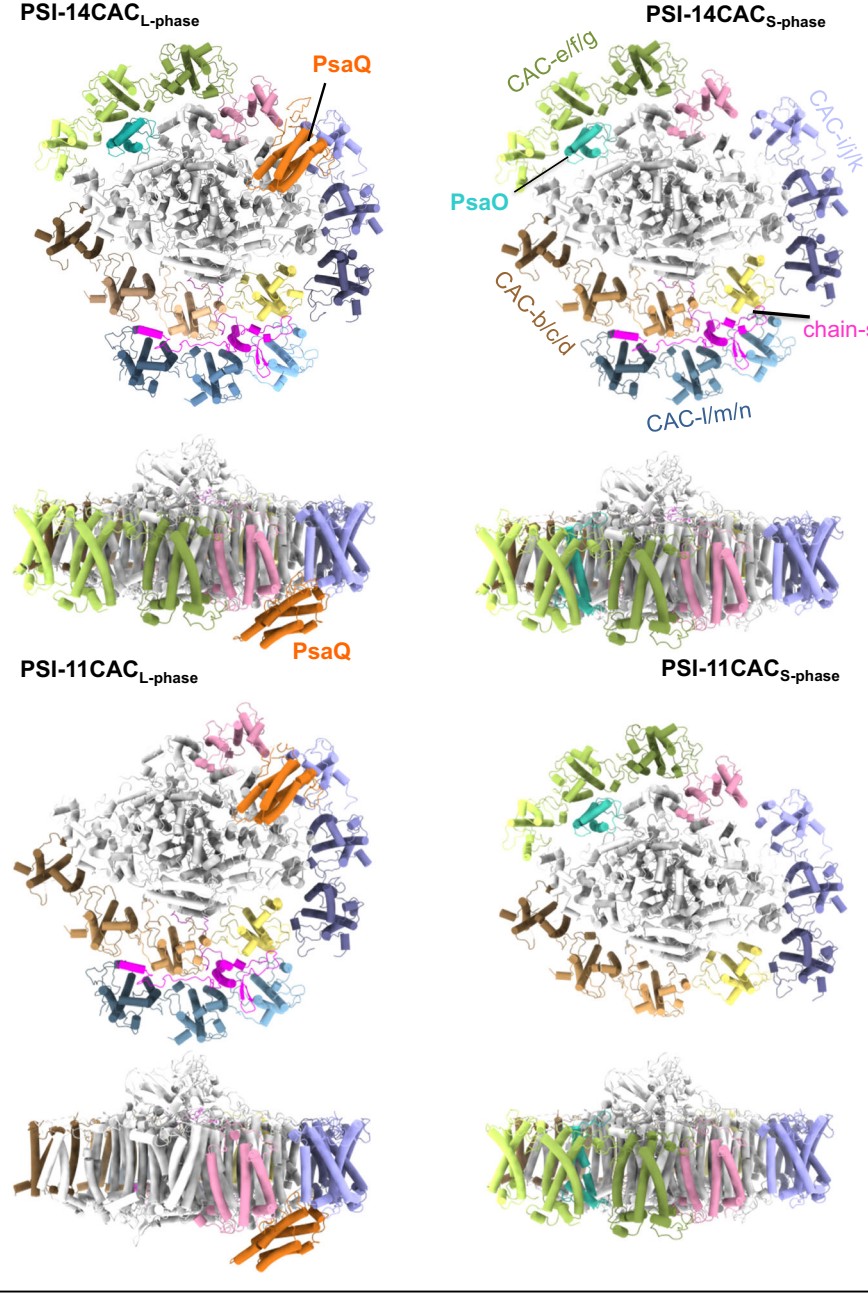

possess a shorter and straight helix C, whereas the other two CACs in these trimers have a longer helix C which is bent at the stromal end. One conserved residue E/Q located at the stromal end of helix C in the second and third CACs is hydrogen-bonded with the residue on the N-terminal loop of the first and second CACs, respectively (Fig. 3a, c). In comparison, this hydrogen bond interaction is lost in the first CACs as the stromal end of their helix C is unwinding. As a result, the first CACs possess a longer AC loop (the loop between helix C and A), which extends toward the interior of the complex, attaching the CAC trimers to the core or the inner CACs (Fig. 3d). Thus, the first CACs in the trimer contribute to the inter-trimer or core-CAC interactions, whereas other two CACs participate in the intra-trimer interactions (Fig. 3d). Interestingly, we found that the second CACs in three CAC trimers (CAC-f/j/m) are identical, whereas CAC-c is a distinct protein. Structural comparison of the two CACs in the second position (CAC-c and CAC-f) showed that one chlorophyll (Chl 605) is present in CAC-f, but absent in CAC-c. Our structural superposition showed that this chlorophyll will clash with the C-terminal helix of chain-s if it binds to

CAC-c (Supplementary Fig. 12). Together, our structural observations suggested that the assembly of a CAC trimer may be initiated at the second CAC, which further recruits the first and third CAC based on their respective structural features. While the first CACs facilitate the assembly of trimers into the complex, they are unable to bind additional CAC, resulting in the trimer formation. Distinct CAC trimers occupy specific positions of the complex, which is likely a result of evolutionary selection.

All CACs and chain-s exhibit high similarities with the corresponding subunits in *Cp*PSI-ACPI structure (Supplementary Table 1, 2), in terms of the protein structure, as well as the pigment composition and arrangement. However, we also observed small structural differences between the two complexes (Supplementary Figs. 13, 14, Supplementary Table 1). For example, compared to the corresponding ACPI (ACPI-3), CAC-b in our structure contains a longer C-terminal tail which harbors an additional chlorophyll (Chl 618). The Chl 618 is positioned in the center of a three-chlorophyll-cluster which is formed by PsaA-805, PsaJ-107 and CAC-c-609 (Supplementary Fig. 14a), thus is pivotal for the efficient energy transfer

**Table 1 | Cryo-EM data collection, refinement and validation statistics**

|  | PSI-14CAC$_{L-phase}$ (EMDB37642) (PDB 8WM6) | PSI-11CAC$_{L-phase}$ (EMDB37654) (PDB 8WMJ) | PSI-14CAC$_{S-phase}$ (EMDB37659) (PDB 8WMV) | PSI-11CAC$_{S-phase}$ (EMDB37660) (PDB 8WMW) |
|---|---|---|---|---|
| **Data collection and processing** |  |  |  |  |
| Magnification | 130,000 | 130,000 | 130,000 | 130,000 |
| Voltage (kV) | 300 | 200 | 300 | 300 |
| Electron exposure (e–/Å$^2$) | 60 | 60 | 60 | 60 |
| Defocus range (μm) | 1.2–2.3 | 1.2–2.2 | 1.2–2.2 | 1.2–2.2 |
| Pixel size (Å) | 1.04 | 1 | 1.04 | 1.04 |
| Symmetry imposed | C1 | C1 | C1 | C1 |
| Initial particle images (no.) | 345,634 | 326,979 | 281,821 | 427,894 |
| Final particle images (no.) | 86,231 | 41,093 | 42,423 | 31,215 |
| Map resolution (Å) FSC threshold | 0.143 | 0.143 | 0.143 | 0.143 |
| Map resolution range (Å) | 2.5-3.7 | 2.6-4.2 | 2.8-4.0 | 3.0–4.2 |
| **Refinement** |  |  |  |  |
| Initial model used (PDB code) | 6LY5, 5ZGB | PSI-14CAC$_{L-phase}$ | PSI-14CAC$_{L-phase}$ | PSI-14CAC$_{L-phase}$ |
| Model resolution (Å) FSC threshold | 0.143 | 0.143 | 0.143 | 0.143 |
| Model resolution range (Å) | 2.7 | 3.0 | 2.9 | 3.3 |
| Map sharpening $B$ factor (Å$^2$) | -57 | -76 | -55 | -76 |
| **Model composition** |  |  |  |  |
| Non-hydrogen atoms | 61,164 | 52,744 | 59,342 | 51,054 |
| Protein residues | 5216 | 4540 | 5017 | 4350 |
| Ligands | 415 | 349 | 407 | 342 |
| ***B* factors (Å$^2$)** |  |  |  |  |
| Protein | 23.85 | 16.29 | 30.52 | 33.63 |
| Ligand | 27.15 | 20.44 | 38.29 | 41.28 |
| **R.m.s. deviations** |  |  |  |  |
| Bond lengths (Å) | 0.008 | 0.011 | 0.011 | 0.010 |
| Bond angles (°) | 1.452 | 1.601 | 1.737 | 1.492 |
| **Validation** |  |  |  |  |
| MolProbity score | 1.85 | 1.82 | 1.69 | 1.73 |
| Clashscore | 7.88 | 9.45 | 6.38 | 7.19 |
| Poor rotamers (%) | 1.93 | 1.88 | 1.48 | 2.21 |
| **Ramachandran plot** |  |  |  |  |
| Favored (%) | 96.82 | 97.48 | 96.65 | 97.65 |
| Allowed (%) | 3.11 | 2.52 | 3.33 | 2.33 |
| Disallowed (%) | 0.08 | 0.00 | 0.02 | 0.02 |

between the PSI core and CAC antennae. Moreover, CAC-n in our structure has one chlorophyll less than the corresponding ACPI (ACPI-9). When we compared the two structures, we found that the absence of this chlorophyll (Chl 614 in ACPI-9) is due to the steric hindrance of the C-terminal tail and one alloxanthin (Alx620) in CAC-n (Supplementary Fig. 14b). These structural differences may reflect the fine-tuning of light harvesting and energy transfer between different cryptophyte species.

**PsaQ structure and interactions with neighboring subunits**
PsaQ contains four helices and harbors two chlorophyll molecules (Fig. 4a). One chlorophyll (Chl a323) is coordinated by the C-terminal residue N234, while the other (Chl a322) is presumably coordinated by a water molecule (Supplementary Fig. 6b). PsaQ possesses a long loop between helix 2 and 3 (2/3-loop) which, together with the two chlorophylls, inserts into the membrane plane at the interfacial region between CAC-i and PsaB (Fig. 4b). Moreover, the N-terminal fragment of PsaQ fills in the gap between CAC-i

and CAC-h (Fig. 4b, c). Multiple hydrogen bond interactions between PsaQ and CAC-i/PsaB (Fig. 4d, e) tether the PsaQ protein to the lumen surface of thylakoid membrane.

While the two L-phase structures contain PsaQ, we did not observe the density corresponding to PsaQ in our two S-phase structures. Moreover, the two *Cp*PSI-ACPI complexes purified from L-phase cells exhibited structures almost identical to our two L-phase PSI-CAC structures, and appeared to contain the lumen-localized PsaQ (Unk1 in *Cp*PSI-ACPI)[12] (Supplementary Fig. 15, Supplementary Table 2). Together, these results suggested that PsaQ is strongly associated with the PSI-CAC complex in L-phase cells, but does not or only loosely bind to the PSI-CAC in S-phase cells.

Several chloroplast proteins with PsbQ-like folding (Supplementary Fig. 9) were found binding to the lumen surface of photosynthetic supercomplexes, and were suggested to facilitate the assembly, folding, stabilization of these supercomplexes[23]. Nevertheless, a pigment-bound PsbQ-like protein has never been identified in phototrophs. PsaQ is unique to

**Fig. 3 | Trimeric organization of CAC.**
**a**, **b** Superposition of CAC apo-proteins (**a**) and chlorophyll (**b**) of four CAC trimers. The black arrow in (**a**) indicates the stromal end of helix C in the first CACs. **c** Specific interaction between neighboring CACs in the trimer, shown in zoom-in views of the regions encircled by boxes in (**a**). **d** Different conformation of helix C and AC loop regions adopted by CACs in four trimers. The longer AC loop in the first CACs and the corresponding region in the second and third CACs are colored in black. The N-terminal fragment of the first and second CACs involved in the interaction with helix C of neighboring CACs are in green color.

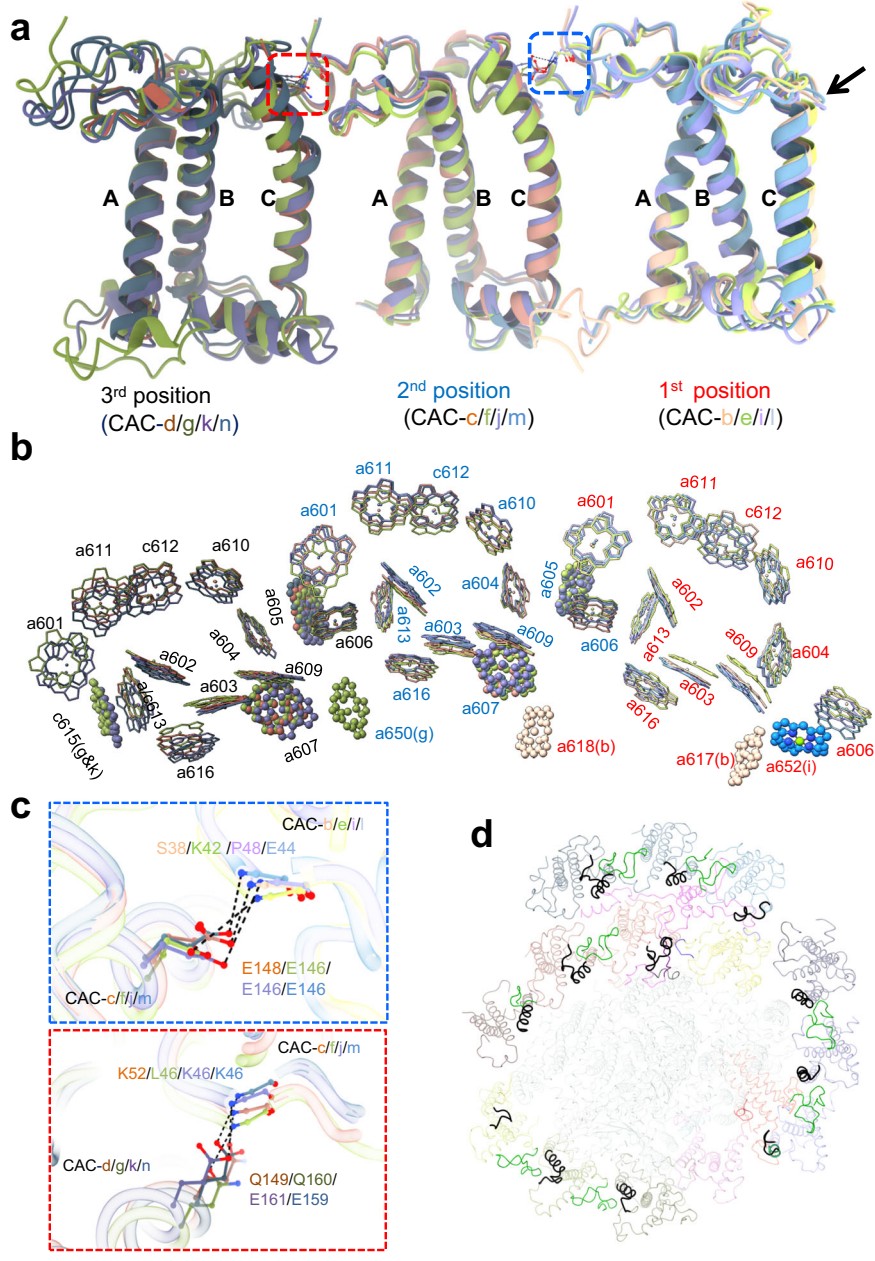

cryptophytes and the chlorophyll-coordinating residue N234 is extremely conserved in PsaQ homologs (Supplementary Fig. 8), suggesting that PsaQ and the bound chlorophylls are crucial for the proper functioning of cryptophyte cells during L-phase. Our structure showed that Chls a322/a323 of PsaQ are in close proximity to Chls a616/a606 in CAC-i and Chl a816 in PsaB (Fig. 4f). These structural observations suggest that in L-phase cells, PsaQ may facilitate efficient excitation energy transfer (EET) in PSI-CAC. In comparison, the fact that PsaQ is absent from (or loosely bound to) PSI-CAC in S-phase cells is in line with the idea that photosynthesis is less efficient in S-phase cells[14,24].

## Structural comparison of *Rs*PSI-CAC with red algal PSI-LHC

Cryptophytes were suggested to be evolutionarily originated from red algae[7,25], which contain LHC (also termed LHCR) proteins serving as the PSI antenna. We thus compared our PSI-CAC structures with two red algal PSI structures that are available, namely PSI-LHCR from *Cyanidioschyzon merolae* (*Cm*PSI-LHCR) and the PSI-LHC moiety in PBS-PSII-PSI-LHC megacomplex from *Porphyridium purpureum* (*Pp*PSI-LHC)[5,26] (Supplementary Fig. 16). We found that CAC-i in our structure

is located more distant from the core and about 9 Å further than the corresponding LHC protein in both red algal PSI structures (Supplementary Fig. 16). In line with this observation, our cryo-EM classification result showed that CAC-i exhibits slightly lower occupancy (ranging from 62% – 90%) in all four PSI-CAC structures. The wider gap between CAC-i and the core in *Rs*PSI-CAC complex is pivotal for the binding of PsaQ (Fig. 4b), which has been evolved only in cryptophytes. These structural observations indicated that PsaQ plays a crucial role in accomplishing cryptophyte-specific functions.

Interestingly, we found that PsaQ and PE545 share several similar features, including their exclusive presence in cryptophytes[11] and their lumenal localization[11,24]. In addition, compared with S-phase cells, PsaQ is more strongly associated with the PSI core (Fig. 2) and PBPs are more abundant (Supplementary Fig. 1b) in L-phase cells[14,18,27]. These data suggest that PsaQ might be co-evolved with PE545 and assist its proper function. Our structural comparison showed that compared with *Cm*PSI-LHCR, *Pp*PSI-LHC is superposed better with the corresponding part of our *Rs*PSI-CAC structures (Supplementary Fig. 16). In addition, previous report suggested that PE545 funnels the excitation energy to both photosystems[28],

**Fig. 4 | PsaQ structure and the interactions with PSI-CAC. a** Overall structure of PsaQ. Four helices are labeled in numbers. The N- and C-terminus (N-ter and C-ter), the loop between helix 2 and 3 (2/3 loop), and two chlorophylls are indicated. **b, c** PsaQ and its chlorophyll molecules insert into the membrane plane between PsaB and CAC-i (**b**) and between CAC-i and CAC-h (**c**). The 2/3 loop and N-terminal fragment of PsaQ are highlighted in black. **d, e** Hydrogen-bond interactions between PsaQ and PsaB (**d**) and between PsaQ and CAC-i (**e**). **f** Closely associated chlorophylls in PsaQ, PsaB and CAC-i. The protein skeleton of CAC-i is omitted for clarity.

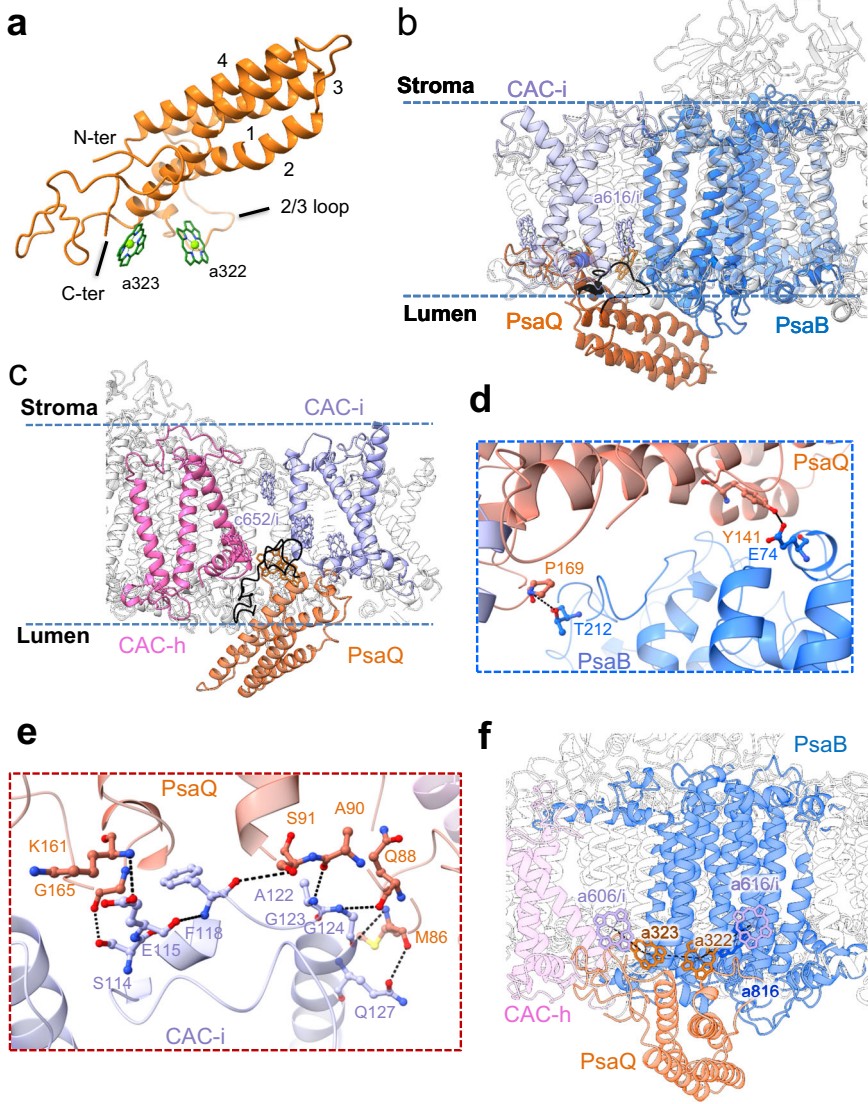

thus we hypothesize that *R. salina* PSII and the PsaQ-bound PSI may assemble into PSI-PSII megacomplex similar to that in *Pp*PBS-PSI-PSI-LHC structure. The potential *Rs*PSI-PSII megacomplex may form a docking site for binding PE545. When we assessed the *Pp*PBS-PSII-PSI-LHC megacomplex structure for features of PSI-PSII association, we found that the PSI interacts with the PSII core through the PsaL-PsaO-PsaK side (Fig. 5a). *Rs*PSI-11CAC$_{L-phase}$ and *Cp*PSI-11ACPI structures also feature an exposed PsaL-PsaO-PsaK side without bound CACs (Supplementary Fig. 15a), implying that a similar PSII-PSI-CAC megacomplex is present in the L-phase cryptophyte cells. We thus superposed our PSI-11CAC$_{L-phase}$ structure onto the PSI part in *Pp*PBS-PSI-PSI-LHC structure, generating a hypothetical *Rs*PSII-PSI-CAC model (Fig. 5a).

Our hypothetic *Rs*PSII-PSI-CAC model showed that the arrangement pattern of PsaQ and the PSII lumenal subunit PsbQ' is almost symmetrical (Fig. 5a). Together with the membrane-embedded CAC-h, these subunits shape a lumenal shallow groove with the width of ~78 Å. The PE545 is a tetramer characterized by a pseudo-two-fold symmetry, with a dimension of ~75 Å × 60 Å × 45 Å[9]. These features enable PE545 to fit into the lumenal groove, resulting in the hypothetical model of *Rs*PE545-PSII-PSI-CAC (Fig. 5b). Nevertheless, it is also possible that PE545 may be located distantly from the thylakoid membranes and the *Rs*PE545-PSII-PSI-CAC is not stable in vivo. To confirm the hypothetic *Rs*PE545-PSII-PSI-CAC model, further experimental evidences are required.

## Discussions

Our structures presented here revealed that cryptophytes possess various types of PSI-CAC complexes. Additionally, the PSI-11CAC complexes exhibit different organization of peripheral antennae at particular growth phases. Moreover, our *Rs*PSI-CAC and previously reported *Cp*PSI-ACPI structures all showed that a cryptophyte-specific protein PsaQ, stably binds to the PSI-CAC complex in L-phase, but is only loosely associated with or even detaches from the PSI-CAC in S-phase cells. Based on our structural data, we speculated that PsaQ may assist the EET of PSI-CAC$_{L-phase}$ complexes, which is in line with the previous report that L-phase cells show higher photosynthetic efficiency[14]. In addition, we also suggested that PsaQ is involved in binding PE545, based on the fact that PsaQ and PE545 share similar characteristics. PE545 is the soluble antenna that situates in the thylakoid lumen of cryptophytes. While previous results suggested that PE545 is able to transfer excitation energy to both photosystems[28,29], precisely where PE545 is positioned relative to PSI and PSII remains unknown. Direct binding of PE545 to the PSII core appears to be sterically hindered by the presence of extrinsic subunits PsbO/PsbU/PsbV and the lumen-extruding domains of PSII core subunits CP43/CP47. Furthermore, the central region of PSI core constitutes the docking site for the lumenal electron donor, which may interfere with the association of PE545 with the PSI core. While direct evidence remains absent, we assume that the *Rs*PSII-PSI-CAC megacomplex similar to the *Pp*PSII-PSI-LHC is present in

**Fig. 5 | The hypothetical *Rs*PE545-PSII-PSI-CAC model. a** The hypothetic *Rs*PSII-PSI-CAC model through superimposing *Rs*PSI-11CAC$_{L-phase}$ structure onto the PSI-LHC moiety of *Pp*PSII-PSI-LHC megacomplex structure. **b** The hypothetic binding site of PBP (represented by PE545, PDB code 1QGW) to the PSII-PSI-CAC megacomplex in cryptophytes.

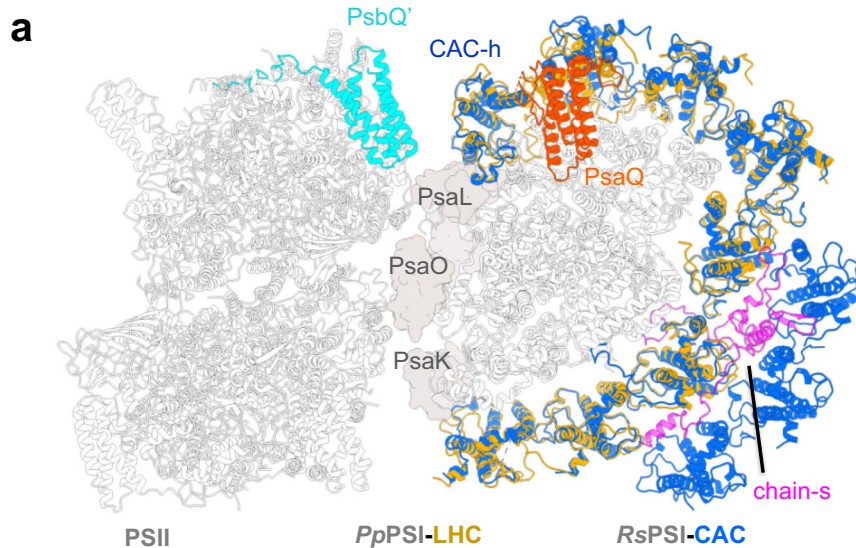

L-phase cryptophyte cells, based on our structural observation that *Rs*PSI-11CAC$_{L-phase}$ and *Pp*PSI-LHC resemble each other, and on the previous suggestion that cryptophytes and red algae are evolutionarily related[8,25]. The *Rs*PSII-PSI-CAC megacomplex and PsaQ may provide a platform for binding lumen-localized PBPs, and chlorophylls in PsaQ and CACs could ensure the excitation energy harvested by PBPs being transferred to the core, as shown by our hypothetical *Rs*PE545-PSII-PSI-CAC megacomplex model. This model is in agreement with an earlier suggestion that PE545 transfers the energy to photosystems through CACs[30,31]. Moreover, our model explains previous data measured by the steady-state and time-resolved fluorescence anisotropy, showing that PBPs funnel the excitation energy to both photosystems with similar efficiency[28,29]. Based on our hypothetic model, the *Rs*PE545-PSII-PSI-CAC megacomplex may also serve to balance the energy distribution between the two photosystems in the *R. salina* cells at the logarithmic growth phase.

Previous results demonstrated that different types of PBPs, namely PE545 and PC645, show highly similar overall structure[9,10]. Thus, a similar megacomplex of the PBP-PSII-PSI-CAC may be ubiquitously present in various cryptophyte species. In addition, we assumed that the proposed PBP-PSII-PSI-CAC megacomplex is more abundant in L-phase cells, as it may meet the energy requirement for the quick growth of L-phase cells. When cryptophytes enter the S-phase, changes of physiological conditions inside the chloroplast, such as lumen pH values, may weaken interactions

between PsaQ and PsaB/CAC-i, resulting in the detachment of PsaQ and the reorganization of the antennae of both photosystems. It is likely that PSI-11CAC$_{L-phase}$ constitutes the preferential form for the potential *Rs*PE545-PSII-PSI-CAC megacomplex formation, whereas other populations of PSI-CAC complexes simultaneously exist in the chloroplast, where they may facilitate the dynamic regulation of light harvesting of PSI under different physiological conditions.

In conclusion, the findings of our study should allow to propose a regulatory mechanism by which *R. salina* adjust the light-harvesting capability and photosynthetic efficiency in line with their growth phase, namely through changing the types of antennae used as well as through rearranging the PSI-associated CACs. Future structural and biochemical assessment of the proposed *Rs*PE545-PSII-PSI megacomplex will provide detailed information required for verifying our hypothesis.

## Materials and method
### Cell culture and growth phase identification
*R. salina* cells (CCMP1319) were obtained from Provasoli-Guillard National Center for Marine Algae and Microbiota (NCMA), and cultured in F/2 medium[32]. The cells were bubbled with air and cultured over a period of 13 days with continuous light of 40–50 μmol photons/m²/s. To identify the growth phase of *R. salina* cells, the number of cells were counted under microscope (Leica DM IL) equipped with 40 x objective lens every day. The

growth curve indicated that *R. salina* cells enter stationary growth phase at the 9th day (Supplementary Fig. 1a).

## Purification and characterization of PSI-CAC complexes

The cells in the logarithmic (cultured for 6 days) and stationary (cultured for 12 days) growth phase were harvested through centrifugation (4500 g, 6 min) separately. Thylakoid membranes were prepared as described by Chua et al. with slight modification[33]. Cells were suspended in buffer A (20 mM HEPES pH 7.5, 10 mM MgCl$_2$·6H$_2$O, 10 mM CaCl$_2$·2H$_2$O, 10 mM NaCl) and broken using a pre-chilled French press at 750 bar. The homogenate was centrifuged at 2000g for 3 min, and the supernatant was further ultra-centrifuged at 70,000 g x 40 min at 4 °C to collect the thylakoid membranes (pellets). To purify the PSI–CAC complex, thylakoid membranes were suspended in buffer A to a final concentration of 0.6 mg/ml in chlorophyll. Dodecyl-α-d-maltoside (α-DDM, 10%) was added to the suspended thylakoids to a final concentration of 1.0%. The mixture was incubated on ice for 15 min, and then centrifuged at 13,800 g, 4 °C for 10 min. The supernatant was loaded on the top of a tube containing buffer A, 0.02% α-DDM and a continuous sucrose density gradient (from 5% to 30%), and fractioned by ultra-centrifugation at 198,200 g for 18 h (Beckman SW41 rotor). The band corresponding to the PSI-CAC complex was divided into the upper half and the lower half, which were collected separately (Supplementary Fig. 1c), and further identified as PSI-11CAC and PSI-14CAC. The PSI-CAC samples were concentrated to about 1.5 mg ml$^{-1}$ in chlorophyll for cryo-EM specimen preparation. The protein composition of PSI-CAC samples used for cryo-EM analysis was identified by sodium dodecylsulfate polyacrylamide gel electrophoresis (SDS-PAGE) and mass spectrometry (Supplementary Fig. 1e).

## Absorption spectra measurement

Absorption spectra of *R. salina* cells and the purified PSI-CAC complexes were measured by UV-Vis spectrometer U3900 (HITACHI). While the content of PBP decreases in *R. salina* cells at stationary growth phase as shown in Supplementary Fig. 1b, the four types of PSI-CAC complexes are almost identical in their absorption spectra (Supplementary Fig. 1d).

## Pigment composition of PSI-CAC complex

Pigment composition was analyzed by high-performance liquid chromatography (HPLC, LC-20AT) apparatus with a prominence fluorescence detector RF20A (Shimadzu, Japan), using a C18 reversed-phase column (Shimadzu, Japan). We assumed that all four types of PSI-CAC complexes have similar pigment composition since they exhibit almost identical absorption spectra. We therefore only analyzed the PSI-14CAC$_{L\text{-phase}}$ sample. The complex sample collected from the sucrose density gradient was concentrated to 1.5 mg ml$^{-1}$ (in chlorophyll) and mixed with 90% (v/v) cold acetone. The mixture was centrifuged at 13,000 g for 10 min to extract pigment molecules[20]. The supernatant containing pigments was loaded on the C18 reversed-phase column (Shimadzu, Japan) and eluted at 20 °C at a flow rate of 1 ml/min with the following steps: 1–20 min, linear gradient of buffer A (acetonitrile: water at the ratio of 85 : 15) from 100% – 0%; 20–23 min, 100% buffer B (ethyl acetate); 23–24 min, linear gradient of buffer B from 100% – 0%; and 24–28 min, 100% buffer A. The eluent was detected at 440 nm with a wavelength detection range of 300–800 nm. Pigments were identified based on their characteristic absorption spectra and retention times of each peak fraction[34,35] (Supplementary Fig. 1f).

## *R. salina* mRNA sequencing

Fresh *R. salina* cells were harvested and stored in liquid nitrogen, then sent to Beijing Genomics Institute (BGI) for high-throughput sequencing analysis. The mRNA extraction, complementary DNA (cDNA) library construction, sequencing and bioinformatics analysis were performed by the technical staffs at BGI. Total RNA was extracted from the fresh *R. salina* cells. The mRNA was enriched by oligo (dT)-attached magnetic beads from the denatured RNA sample with the opened secondary structure. Then the mRNAs were fragmented and reversely transcript to cDNA fragments,

which were further amplified by PCR. The PCR products were denatured to obtain single-stranded DNA products, which were then cyclized and used for sequencing. A total of 10.14 Gb data were sequenced in this project. The raw data was filtered with SOAPnuke (v1.6.5)[36] to obtain clean reads, which was then assembled by Trinity v2.0.6[37]. Clean data were mapped to the assembled unique gene by Bowtie2 v2.2.5[38], and the expression level of gene was calculated by RSEM (v1.3.1)[39]. Gene annotation was performed using seven major functional databases (NR, NT, SwissProt, KOG, KEGG, GO and Pfam). Two parallel tests yielded the same results.

## Grid preparation and cryo-EM data acquisition

The PSI-CAC samples were added to a glow discharged holey carbon grid (GIG-A31213). The sample was vitrified by flash plunging the grid into liquid ethane using vitrobot Mark IV (FEI) with blotting time of 4 s, force level of 4 and humidity of 100%. The micrographs were collected using SerialEM v3.6/3.6.1 data collection software on 300 kV Titan Krios (FEI) microscope equipped with K2 direct electron detector (Gatan) (for PSI-14CAC$_{L\text{-phase}}$, PSI-11CAC$_{S\text{-phase}}$ and PSI-14CAC$_{S\text{-phase}}$) and 200 kV Talos Arctica (FEI) equipped with K2 direct electron detector (Gatan) (for PSI-11CAC$_{L\text{-phase}}$). The parameters for data collection were summarized in Table 1.

## Data processing

All movie stacks were corrected by program Motion Cor 2 with dose weighting[40]. CTF parameters of each movie were estimated by Gctf[41]. Images were processed using the program relion 3.1[42]. Automatic particle picking and reference-free 2D classifications were performed. The selected particles were used as templates for further picking procedure, then the picked particles were applied to 2D and 3D classification. After 3D non-uniform refinement and sharpening, CTF refinement (global and local) and post processing were performed. The data collection and processing procedures were summarized in Supplementary Fig. 2–5. The final overall resolutions of maps of *R. salina* PSI-14CAC$_{L\text{-phase}}$, PSI-11CAC$_{L\text{-phase}}$, PSI-14CAC$_{S\text{-phase}}$, PSI-11CAC$_{S\text{-phase}}$ are 2.7 Å, 3.0 Å, 2.9 Å and 3.3 Å, respectively. Local refinement targeting the PsaQ in PSI-14CAC$_{L\text{-phase}}$ was performed, resulting in a final local map with the resolution of 3.5 Å. The resolutions of the final maps were calculated using ResMap[43].

## Model building, refinement and validation

For model building of the PSI-14CAC$_{L\text{-phase}}$, the PSI core from the cryo-EM structure of diatom PSI-FCPI (PDB code: 6LY5) was manually docked into the 2.7 Å resolution cryo-EM map using UCSF Chimera[44]. The CAC antennae were built by docking a single LHCR structure from the red algal PSI-LHCR model (PDB code 5ZGB) into the cryo-EM map. The identification of each individual CAC antenna was based on the best match of the specific sequence with the cryo-EM densities. The amino acid sequences of CAC antennae were then mutated to their counterparts in *R. salina*. To identify the PsaQ protein, we first traced a short peptide containing 10 residues ($^{1}$KQRVVLAGKI$^{10}$) at the local helix region based on the 3.5 Å local map. We then searched the *R. salina* protein sequences derived from our transcriptome sequencing result, and found that a fragment of PsaQ ($^{95}$KQRTVLAGKI$^{104}$) shows high similarity with the short peptide (Supplementary Fig. 6c). Based on the determined amino acid sequence of PsaQ, we manually built the PsaQ model and found that most amino acids of PsaQ fit the density well (Supplementary Fig. 6b). After the model building of PSI-14CAC$_{L\text{-phase}}$ structure, automatic real-space refinements in Phenix[45] and manual correction in COOT[46] were carried out iteratively. The models of PSI-11CAC$_{L\text{-phase}}$, PSI-14CAC$_{S\text{-phase}}$ and PSI-11CAC$_{S\text{-phase}}$ were built by docking the corresponding part of refined PSI-14CAC$_{L\text{-phase}}$ model into the cryo-EM maps, and then performing manual adjustment and real-space refinements. The geometries of four structures were assessed using Phenix[45] and MolProbity[47] and detailed information were listed in Table 1. The structural figures were prepared using ChimeraX 1.2[44] (Molecular Graphics System). The figure of sequence alignment was generated using ESPript65 v.3.0.

## Statistics and reproducibility

The counting of *R. salina* cell numbers was performed in three biological replicates. Other experiments, including the absorption spectra measurement, SDS-PAGE and HPLC analysis were performed in 2-6 biological replicates, with reproducible results obtained.

## Reporting summary

Further information on research design is available in the Nature Portfolio Reporting Summary linked to this article.

## Data availability

The atomic coordinates and cryo-EM density maps of PSI-CAC complexes have been deposited in the Protein Data Bank (PDB) and Electron Microscopy Data Bank (EMDB) with accession codes: 8WM6 and EMDB-37642 for PSI-14CAC$_{L-phase}$ structure, 8WMJ and EMDB-37654 for PSI-11CAC$_{L-phase}$ structure, 8WMV and EMDB-37659 for PSI-14CAC$_{S-phase}$ structure, 8WMW and EMDB-37660 for PSI-11CAC$_{S-phase}$ structure. The locally refined cryo-EM map of PsaQ has been deposited in the EMDB with accession code EMDB-37674. RNA sequencing data have been deposited into the NCBI Sequence Read Archive (SRA) under the BioProject ID PRJNA1104392. The uncropped gel image of Supplementary Fig. 1e is provided in Supplementary Fig. 17. All other data generated or analyzed in this study is available from the corresponding author on reasonable request.

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

## Acknowledgements
We thank B. Zhu, X. Huang, X. Li and other staff members at the Center for Biological Imaging (IBP, CAS) for their support in data collection; L. Niu for mass spectrometry; L. Shi for assisting pigment identification. We thank Torsten Juelich (University of Chinese Academy of Sciences) for linguistic assistance during the preparation of the article. The project is funded by the Strategic Priority Research Program of CAS (XDB37020101), National Natural Science Foundation of China (31930064, 32171183), the CAS Project for Young Scientists in Basic Research (#YSBR-015) and Youth Innovation Promotion Association, Chinese Academy of Sciences (Y2022038).

## Author contributions
M.L. conceived and coordinated the project; L.S. performed the sample preparation, data collection, and reconstructed the cryo-EM maps; S.Z. built and refined the structure models; X.S. helped in cryo-EM data collection; X.Z. and X.A. assisted in *R. salina* cell culturing and complex isolation; S.Z., L.S. and M.L. analyzed the structures; S.Z. and M.L. wrote the manuscript; all authors discussed and commented on the results and the manuscript.

## Competing interests
The authors declare no competing interests.
