## [Peer review file · Communications Biology]

Reviewers' comments:

Reviewer #1 (Remarks to the Author):

The paper by Zhang present four photosystem I structures obtained from the cryptophytes alga *Rhodomonas salina* in either the stationary or log phase of growth. This is the second example of a PSI structure from the cryptophytes published after a structure from *Chroomonas placoidea*. This manuscript contains a few interesting additions, first is the identification of PsaQ which was not identified in the previous manuscript, second is the potential finding that PsaQ is associated with PSI during log phase only. In my opinion these findings, together with this being only the second structure of PSI from this interesting group, is sufficient for publication in *Communications Biology*, I find that the presentation of the data and the discussion of its implication can and should be greatly improved. There are some issues with data processing, its presentation and model construction. I have listed them below and they should be corrected before resubmission.

Since the main conclusion of the paper revolve around the difference in PsaQ association, any information the authors can further preset on this finding will strengthen the manuscript greatly. The identification of PsaQ should be added to the supplementary information, the authors state that a short amino acid sequence was identified from the map, the hits obtained from that sequence should be shown together with the map corresponding to the query sequence used, the authors should try and estimate the confidence in the identification. Alternatively, the authors can supply additional evidence for the presence of PsaQ in the complex such as MS of other means of identification. The authors present pigment analysis for only one of their samples. The absorption spectra for the different purified complexes are not presented. This should be presented at least for comparing L vs S complexes. If the difference spectra suggest any changes in pigment composition, this should be analyzed by HPLC.

The reasoning for using different names than the ones used in Zhao et al is not clear. With the exception of PsaQ which was named unk1 by Zhao et al, I don't see why CAC's should not be named as they were in Zhao et al. Same goes for chain S, which was named ACPI-S by Zhao et al. why add confusion to the already messy antennae naming world.

The authors (and Zhao et al) describe the organization of CACs/ACPs are sets of trimers. In all the current maps the region occupied by chain i seem to be significantly worse, the authors should determine the occupancy of chain i using either classification or refinement and include this data in the manuscript.

Supplementary figures on data processing should include the identity of the software used in each step or series of steps. For example, in figure S2, (motion correction 2.1 ?). When 3D classification was used the percent of each class should be noted.

Presumably none of the datasets contained pure PSI-11CAC/PSI-14CAC these classes should be identified in the data processing work flow.

Especially in the case of supplementary figure 2 and 4 the resolution of the final reconstruction may benefit from including all good PSI classes in the steps prior to separating out PSI-11CAC.

Figure 4b, CAC-I or CAC-I naming of all chains/CAC's should be consistence throughout the manuscript.

The authors present RedCap as an LHC/CAC/ACP in some figures and in others I think it is labeled using its chain id (h).

In line 113 the authors use figure 3 as a reference of RedCap, I could barely find any mention for RedCap in figure 3 and while figure 4 shows RedCap, it does not show pigments.

PsaQ focused map – the map itself appear to be of good quality. The assignment of Chl324 seems questionable to me, especially with a coordinated water residue which cannot be supported at the resolution. The authors should provide more convincing maps if they was to keep this assignment.

Three of the pdb files contained CLA atoms that were grossly misplaced, these are PSI-11CAC-S and PSI-11CAC-L which contain the following atoms from CLA820 C15 and CLA830 C8,9,10,11 both from Chain B are placed outside the membrane plane in a clearly erroneous position and PSI-14CAC-S which contains CLA820 C15 from chain B. The authors must supply the files they uploaded to the PDB

and EMDB for review after the removal of these errors.

Line 202 – the authors claim that PsaQ has higher stability or abundance in L phase, but they haven't showed anything about the cellular state of PsaQ. No information is presented about its abundance or stability in cells. All statements regarding this fact should be corrected or alternatively more data regarding the cellular abundance and half life of PsaQ should be presented.

The first sentence of the discussion is somewhat misleading (line 227). I don't think the authors found that "Our structures presented here revealed that cryptophyte PSI binds CACs in different arrangement at particular growth phases". If anything, they found the opposite, that the arrangement of CAC's is not affected by the growth phase and most changes are found at the level of PsaQ binding. On line 231 the authors state "Based on our structural data, we suggested that PsaQ assists the EET of PSI-CACL-phase complexes, which is in line with the previous report that L-phase cells show higher photosynthetic efficiency" two papers having nothing to do with stationary phase vs log phase growth are cited, the authors may have meant to cite Funk on this matter. The only information the author present regarding light harvesting in cells can be found in Supp figure 1b. It does suggest that there is a decrease in PE545 levels in S cells, but it does not offer any data on PSI or PSII transfer. The authors may present such data, such as 77 K emission, if they want to further substantiate their proposed PE545/PsaQ model. while it is acceptable to include speculative models in the discussion section, I find the phrasing of the current discussion not sufficiently critical and lacking in terms of correct citations.

Reviewer #2 (Remarks to the Author):

The manuscript by Zhang and co-workers reports four cryo-EM structures of the photosystem I–light-harvesting supercomplex of cryptophyte algae obtained at different growth phases. The first structure for this PSI-LHC supercomplex was already obtained for a different cryptophyte species by another group last year, showing interesting differences with other algal groups like red algae or diatoms. The present paper, however, reports a very important novelty compared to that previous structure: the identification of a cryptophyte-unique protein PsaQ, carrying three pigments, which is only present in the supercomplex obtained from algae grown at the logarithmic growth phase. Indeed, that protein subunit was observed in the previously published structure, but with low resolution, so its amino acids or pigments could not be assigned.

The importance of this finding relies on the fact that PsaQ faces the lumen, where the unique water-soluble phycobiliprotein (PBP) antenna complexes that characterize cryptophytes are located, and its pigment composition reported here suggests it can be the key complex connecting PBPs with PSI. Moreover, it can be key in regulating the photosynthetic efficiency, thus explaining why it is only present in the logarithmic growth phase and not in the stationary phase. The authors also provide convincing evidence of a possible arrangement of the PSI-PSII-PBP arrangement that would connect energy transfer from PBP antennas to both photosystems. This findings are very exciting for the wide community working in photosynthesis, and of key relevance for researchers studying cryptophytes, as they pave the way for a global analysis of the light harvesting mechanisms and regulation in cryptophytes. Overall the conclusions are well-supported by the data, the methods are appropriate and well-described, and I don't have any criticism or further point to ask the authors. Therefore, I strongly recommend publication of this excellent article in *Communications Biology* in its current form.

Reviewer #3 (Remarks to the Author):

Photosynthetic cryptophytes evolved from red algae by secondary endosymbiosis. Cryptophyte algae are single-cellular organisms that go through logarithmic and stationary growth phase, in which cryptophytes have different antenna systems. The structure of PSI-ACPI from *Chroomonas placoidea* has been solved and indicated the assembly and pigment arrangement of LHCs in cryptophyte PSI at

L-phase, while the details of how the antennae of PSI are organized at S-phase still unclear.

In this manuscript, the authors describe four cryo-electron microscopy structures of PSI-CACs from *Rhodomonas salina* at L- and S-phase. The structural comparison reveals how cryptophytes adjust their PSI antennae in response to the different growth state. The results provide a solid structural basis for unraveling the mechanisms of regulatory mechanism of photosynthetic organisms to cope with the environmental stresses. Nevertheless, some elements of the present manuscript, as shown below, need to be further revised before publication.

1. About the Fig.2:

The presence of the four-helix-bundle protein (PsaQ) is the only difference between two PSI-14CAC structures. The authors should highlight it in Fig.2.

2. Line 94: The overall folding of this protein is similar to the PSII extrinsic subunit PsbQ (Fig. S9); we therefore termed it PsaQ.

1). The authors show the different structures of PsbQ in Fig.S9, it might be better if the structures of PsaQ and PsbQ are aligned;

2). Although the feature of the four-helix-bundle protein (named as PsaQ here) is similar to PsbQ, this feature is general in PSII including PsbQ, Psb31, Psb27., etc. Thus, the naming of this subunit still needs further consideration.

3. About the RedCAP subunit:

The RedCAP is also found in PpPSI-LHC, and the binding location seems to be similar. It would be helpful if the authors could compare the structures of RedCAP subunits between cryptophyte PSI and the red algal PSI-LHC.

4. About the PsaQ subunit:

As cryptophytes are evolutionarily originated from red algae. Structurally, the PsaQ is similar to PsbQ. Therefore, is the PsaQ genetically derived from some subunits in red algae or cyanobacteria?

5. About the hypothetic RsPE545-PSII-PSI-CAC model:

The antennae PE545 are densely stacked in the lumen, and PSII and PSI are embedded into the membranes. The authors proposed a hypothetic RsPE545-PSII-PSI-CAC model according to the dimension of the luminal groove (Fig.5). However, it is likely that the most of PE545 are located away from the membranes, where the PE545 does not seem to bind to PSII and PSI that on the membranes. Thus, the RsPE545-PSII-PSI-CAC may not stable in native content.

6. The PsaQ is associated/disassociated to PSI core under L- and S-phase, and it interacts with CAC-I and PsaB. In addition to providing an efficient energy transfer pathway, will the absence of PsaQ affect changes in the overall structure of PSI core? The authors may be able to make a detailed comparison of related pigments and protein structures in L- and S-phase.

Response to Reviewers' comments:

Reviewer #1 (Remarks to the Author):

The paper by Zhang present four photosystem I structures obtained from the cryptophytes alga *Rhodomonas salina* in either the stationary or log phase of growth. This is the second example of a PSI structure from the cryptophytes published after a structure from *Chroomonas placoidea*. This manuscript contains a few interesting additions, first is the identification of PsaQ which was not identified in the previous manuscript, second is the potential finding that PsaQ is associated with PSI during log phase only. In my opinion these findings, together with this being only the second structure of PSI from this interesting group, is sufficient for publication in *Communications Biology*, I find that the presentation of the data and the discussion of its implication can and should be greatly improved.

There are some issues with data processing, its presentation and model construction. I have listed them below and they should be corrected before resubmission.

Response: Thank you for your positive comments and valuable suggestions. We have corrected the models and modified the manuscript accordingly.

(1) Since the main conclusion of the paper revolve around the difference in PsaQ association, any information the authors can further preset on this finding will strengthen the manuscript greatly. The identification of PsaQ should be added to the supplementary information, the authors state that a short amino acid sequence was identified from the map, the hits obtained from that sequence should be shown together with the map corresponding to the query sequence used, the authors should try and estimate the confidence in the identification. Alternatively, the authors can supply additional evidence for the presence of PsaQ in the complex such as MS of other means of identification.

Response: Thanks for your suggestion. To determine the identity of PsaQ, we first obtained the protein sequence library of *R. salina* based on our transcriptome sequencing analysis. Next, we checked the region of the cryo-EM density with good quality, and tentatively built a short peptide containing 10 amino acid residues (¹KQRVVLAKGI¹⁰) according to our cryo-EM map (Response Fig. 1a). We then searched our *R. salina* protein sequence library for homologous sequence of this short

peptide. We found that the sequence (⁹⁵KQRTVLAGKI¹⁰⁴) of one unidentified protein, which we designated PsaQ later, shows high homology (Response Fig. 1b). Next, we used the online server XtalPrediction to predict the secondary structure of PsaQ based on its amino acid sequence. The result showed that PsaQ is mainly composed of α -helices (Response Fig. 1c) which is consistent with the feature of cryo-EM density. We thus built the model of the entire protein using the sequence of PsaQ, and found that the sequence and the cryo-EM density matched well (Supplementary Fig. 6b in the revised manuscript). In addition, the structure of PsaQ superposes well with the structure predicted by DDMFOLD except the terminal regions (Response Fig. 1d). Furthermore, we also loaded the PSI-CAC complexes purified from L-phase cells for SDS-PAGE analysis, and cut the gel in the area around the 25 kDa protein marker for mass spectrometry (LC-MS/MS), as the theoretical molecular weight of PsaQ is ~24 kDa. Although we have not yet determined which band corresponds to PsaQ exactly, our results confirmed the presence of PsaQ in our PSI-CAC complexes purified from L-phase cells (Response Fig. 1e). Based on these evidences, we can confidently identify PsaQ in our structures. We have provided the detailed informaton of PsaQ identification in the Method and in Supplementary Figure 6c in our revised manuscript.

Response Fig. 1. Identification of PsaQ.

a. The fitting of the cryo-EM density with the short peptide ($^1KQRVVLAKGI^{10}$) we originally built (carbon atoms shown in yellow) and with the PsaQ sequence ($^{95}KQRTVLAKGI^{104}$) from the final model (carbon atoms shown in green). *b.* The blast result of the short peptide, showing high identity with the sequence of *R. salina* PsaQ. *c.* Secondary structure of PsaQ predicted by the online server XtalPrediction. *d.* Superposition of the experimentally built PsaQ structure (orange) with the model predicted by DDMFOLD according to the sequence of PsaQ (blue). *e.* The LC-MS/MS result. Red box indicates the piece of gel (shown in the left) used for mass spectrometry. The detected proteins (shown in the right) with the highest scores are CAC-b/h/g/d/i, chain-s, a hypothetical protein and PsaQ (highlighted in red).

(2) The authors present pigment analysis for only one of their samples. The absorption spectra for the different purified complexes are not presented. This should be presented at least for comparing L vs S complexes. If the difference spectra suggest any changes in pigment composition, this should be analyzed by HPLC.

Response: Thank you for this important suggestion. We have measured the absorption spectra of all four samples, and found that they are highly similar (Response Fig. 2). The results are also shown in Fig. S1d in our revised manuscript.

Response Fig. 2. Absorption spectra of four types of PSI-CAC complex.

(3) The reasoning for using different names than the ones used in Zhao et al is not clear. With the exception of PsaQ which was named unk1 by Zhao et al, I don't see why CAC's should not be named as they were in Zhao et al. Same goes for chain S, which was named ACPI-S by Zhao et al. why add confusion to the already messy antennae naming world.

Response: Thank you very much for pointing this out and giving us an opportunity to explain. The reason that we used the name "chain-s" instead of "ACPI-S" is as

following. ACP represents the alloxanthin-chlorophyll a/c binding protein which usually contains three transmembrane helices and binds alloxanthin molecules. However, chain-s/ACPI-S contains only one transmembrane helix and binds no alloxanthin based on both our structures and the structures reported by Zhao et al (Response Fig. 3). We thus designated this protein “chain-s” to distinguish it from common light-harvesting proteins CAC/ACPI. For the name of antenna proteins, we do realize that ACPI is more accurate name for cryptophyte PSI antennae. However, CAC is the most commonly used name for the membrane-embedded light-harvesting proteins in cryptophytes in literatures. In addition, the name “ACPI” appears to imply that these proteins are the antennae of the PSI but not of the PSII, since “LHCI” and “LHCII” in green lineage represent antenna proteins bound to the PSI core and the PSII core respectively. However, this may not be the case in cryptophytes, since we found that the PSI and PSII in *R. salina* share at least one CAC protein (our unpublished data). For these reasons, we hope for your understanding that we wish to keep the names of “CAC” and “chain-s” in our manuscript. Nevertheless, we do realize that using different names for the homologues protein will lead to inconvenience and sometimes confusion for readers, thus we have added a Supplementary table (Table S3) in our revised manuscript to summarize the names of chain-s/ACPI-S, PsaQ/Unk1 and all CACs/ACPIs.

Response Fig. 3. Structures of chain-s (magenta) and ACPI-S (grey-blue). Chl a, Chl c, a-carotene and crocoxanthin molecules are shown in green, yellow, brown and grey-blue respectively.

(4) The authors (and Zhao et al) describe the organization of CACs/ACPs are sets of trimers. In all the current maps the region occupied by chain i seem to be significantly worse, the authors should determine the occupancy of chain i using either classification or refinement and include this data in the manuscript.

Response: Thank you for your suggestion. We calculated the occupancy of chain i (CAC-i) in four types of PSI-CAC using no alignment 3D classification of the final set of particles (Response Fig. 4). The results showed that CAC-i indeed exhibits lower occupancy (ranging from 62% to 90%). Our structural comparison of cryptophyte PSI-CAC with red algal PSI-LHCR indicate that CAC-i shifts away from the PSI core for about 9 Å in cryptophyte compared with the corresponding subunit in red algae (Fig. S16 in the revised manuscript), this could be one of the reasons that CAC-i exhibits weaker association with the PSI core. We have included this data in our revised manuscript (Line 195).

Response Fig. 4. The occupancy of CAC-i in four types of PSI-CAC complexes.

(5) Supplementary figures on data processing should include the identity of the software used in each step or series of steps. For example, in figure S2, (motion correction 2.1 ?). When 3D classification was used the percent of each class should be noted.

Response: We are sorry for the typo and thank you for pointing this out. We have modified Figs. S2-S5 to include the identity of the software and provided the percentage of each class in 3D classification in the revised manuscript.

(6) Presumably none of the datasets contained pure PSI-11CAC/PSI-14CAC these classes should be identified in the data processing work flow. Especially in the case of supplementary figure 2 and 4 the resolution of the final reconstruction may benefit from including all good PSI classes in the steps prior to separating out PSI-11CAC.

Response: Thank you for your suggestion. It is indeed that each dataset contains particles of other types of PSI-CAC complex. This is because that the PSI-11CAC and PSI-14CAC together exhibit one wide band after the sucrose density gradient centrifugation (Fig. S1c), thus cannot be separated very well. Our 3D-classification results showed that no good particles of other types of PSI-CAC are present in the two PSI-14CAC datasets. However, other types of PSI, including PSI-14CAC and PSI-8CAC, were all observed in the two PSI-11CAC datasets (Figs. S2-5 in the revised manuscript). Therefore, we reconstructed the two PSI-11CACs by including PSI particles from other classes as the reviewer suggested. However, the results showed that no significant improvement in resolution was achieved (3.0 and 3.3 Å (before) vs 3.16 and 3.26 Å (after)). We assumed that other PSI classes contain some bad particles which may impair the resolution of the final reconstruction.

Response Fig. 5. 3D-classification of PSI-11CAC_{L-phase} and PSI-11CAC_{S-phase}.

(7) Figure 4b, CAC-I or CAC-I naming of all chains/CAC's should be consistent throughout the manuscript.

Response: Thank you for your suggestion. We have modified Figure 4b and checked other figures as well as the text throughout the manuscript to ensure consistent naming.

(8) The authors present RedCap as an LHC/CAC/ACP in some figures and in others I think it is labeled using its chain id (h).

Response: Thank you for your suggestion, we have modified the text and figures in our revised manuscript, using the unified name "CAC-h" in most cases.

(9) In line 113 the authors use figure 3 as a reference of RedCap, I could barely find any mention for RedCap in figure 3 and while figure 4 shows RedCap, it does not show pigments.

Response: Thank you for pointing out this mistake. It is Figure S11 that shows the differences of protein conformation and pigment number between RedCap and other CACs. We have cited the correct figure as following "*It is noteworthy that CAC-h constitutes a RedCap (Engelken, Brinkmann et al. 2010) (red lineage chlorophyll a/b-binding-like protein), which exhibits slightly different conformation and binds fewer pigment molecules compared to canonical CACs which possess 11-15 chlorophylls and five carotenoids (Fig. S11, Table S2)*" in the revised manuscript.

(10) PsaQ focused map – the map itself appear to be of good quality. The assignment of Chl324 seems questionable to me, especially with a coordinated water residue which cannot be supported at the resolution. The authors should provide more convincing maps if they was to keep this assignment.

Response: Thank you very much for your valuable suggestion. We agree with the reviewer that the density of Chl 324 is not good enough to allow the confident assignment of the chlorophyll. We therefore removed this chlorophyll in our final models, and modified the manuscript accordingly.

(11) Three of the pdb files contained CLA atoms that were grossly misplaced, these are PSI-11CAC-S and PSI-11CAC-L which contain the following atoms from CLA820 C15 and CLA830 C8,9,10,11 both from Chain B are placed outside the membrane plane

in a clearly erroneous position and PSI-14CAC-S which contains CLA820 C15 from chain B. The authors must supply the files they uploaded to the PDB and EMDB for review after the removal of these errors.

Response: Thank you for pointing out the mistake in our models. We have double checked and corrected all our PDB files. The revised PDB and EMDB files have been uploaded through the online portal.

(12) Line 202 – the authors claim that PsaQ has higher stability or abundance in L phase, but they haven't showed anything about the cellular state of PsaQ. No information is presented about its abundance or stability in cells. All statements regarding this fact should be corrected or alternatively more data regarding the cellular abundance and half life of PsaQ should be presented.

Response: We are sorry for the unclear description. We have modified the sentence as following *“Interestingly, we found that PsaQ and PE545 share several similar features, including their exclusive presence in cryptophytes (Ludwig and Gibbs 1989) and their luminal localization (Ludwig and Gibbs 1989, Kana, Prasil et al. 2009). In addition, compared with S-phase cells, PsaQ is more strongly associated with the PSI core (Fig. 2) and PBPs are more abundant (Fig. S1b) in L-phase cells (Cheregi, Kotabova et al. 2015, Heidenreich and Richardson 2020, Yamamoto, Bossier et al. 2020). These data suggest that PsaQ might be co-evolved with PE545 and assist its proper function.”*.

(13) The first sentence of the discussion is somewhat misleading (line 227). I don't think the authors found that “Our structures presented here revealed that cryptophyte PSI binds CACs in different arrangement at particular growth phases”. If anything, they found the opposite, that the arrangement of CAC's is not affected by the growth phase and most changes are found at the level of PsaQ binding.

Response: Thank you for pointing out the inaccurate statement. We have rephrased the sentence as following *“Our structures presented here revealed that cryptophytes possess various types of PSI-CAC complexes. Additionally, the PSI-11CAC complexes exhibit different organization of peripheral antennae at particular growth phases. Moreover, our RsPSI-CAC and previously reported CpPSI-ACPI structures all showed that”*.

(14) On line 231 the authors state “Based on our structural data, we suggested that PsaQ

assists the EET of PSI-CACL-phase complexes, which is in line with the previous report that L-phase cells show higher photosynthetic efficiency” two papers having nothing to do with stationary phase vs log phase growth are cited, the authors may have meant to cite Funk on this matter. The only information the author present regarding light harvesting in cells can be found in Supp figure 1b. It does suggest that there is a decrease in PE545 levels in S cells, but it does not offer any data on PSI or PSII transfer. The authors may present such data, such as 77 K emission, if they want to further substantiate their proposed PE545/PsaQ model. while it is acceptable to include speculative models in the discussion section, I find the phrasing of the current discussion not sufficiently critical and lacking in terms of correct citations.

Response: Thank you for your important suggestion. We have modified the sentence and corrected the citation in the revised manuscript.

Reviewer #2 (Remarks to the Author):

The manuscript by Zhang and co-workers reports four cryo-EM structures of the photosystem I–light-harvesting supercomplex of cryptophyte algae obtained at different growth phases. The first structure for this PSI-LHC supercomplex was already obtained for a different cryptophyte species by another group last year, showing interesting differences with other algal groups like red algae or diatoms. The present paper, however, reports a very important novelty compared to that previous structure: the identification of a cryptophyte-unique protein PsaQ, carrying three pigments, which is only present in the supercomplex obtained from algae grown at the logarithmic growth phase. Indeed, that protein subunit was observed in the previously published structure, but with low resolution, so its amino acids or pigments could not be assigned. The importance of this finding relies on the fact that PsaQ faces the lumen, where the unique water-soluble phycobiliprotein (PBP) antenna complexes that characterize cryptophytes are located, and its pigment composition reported here suggests it can be the key complex connecting PBPs with PSI. Moreover, it can be key in regulating the photosynthetic efficiency, thus explaining why it is only present in the logarithmic growth phase and not in the stationary phase. The authors also provide convincing evidence of a possible arrangement of the PSI-PSII-PBP arrangement that would connect energy transfer from PBP antennas to both photosystems. This findings are very exciting for the wide community working in photosynthesis, and of key relevance for researchers studying cryptophytes, as they pave the way for a global analysis of the

light harvesting mechanisms and regulation in cryptophytes. Overall the conclusions are well-supported by the data, the methods are appropriate and well-described, and I don't have any criticism or further point to ask the authors. Therefore, I strongly recommend publication of this excellent article in *Communications Biology* in its current form.

Response: We greatly appreciate the reviewer for the encouragements and appreciation of our work.

Reviewer #3 (Remarks to the Author):

Photosynthetic cryptophytes evolved from red algae by secondary endosymbiosis. Cryptophyte algae are single-cellular organisms that go through logarithmic and stationary growth phase, in which cryptophytes have different antenna systems. The structure of PSI-ACPI from *Chroomonas placoidea* has been solved and indicated the assembly and pigment arrangement of LHCs in cryptophyte PSI at L-phase, while the details of how the antennae of PSI are organized at S-phase still unclear.

In this manuscript, the authors describe four cryo-electron microscopy structures of PSI-CACs from *Rhodomonas salina* at L- and S-phase. The structural comparison reveals how cryptophytes adjust their PSI antennae in response to the different growth state. The results provide a solid structural basis for unraveling the mechanisms of regulatory mechanism of photosynthetic organisms to cope with the environmental stresses. Nevertheless, some elements of the present manuscript, as shown below, need to be further revised before publication.

Response: We thank the reviewer for the positive comments and important suggestions. We have modified our manuscript accordingly.

1. About the Fig.2: The presence of the four-helix-bundle protein (PsaQ) is the only difference between two PSI-14CAC structures. The authors should highlight it in Fig.2.

Response: Thank you for your suggestion. We have modified Fig. 2 in the revised manuscript accordingly.

2. Line 94: The overall folding of this protein is similar to the PSII extrinsic subunit PsbQ (Fig. S9); we therefore termed it PsaQ.

1). The authors show the different structures of PsbQ in Fig.S9, it might be better if the structures of PsaQ and PsbQ are aligned;

Response: Thanks for your suggestion. We have modified Fig. S9 to show the structural superposition of our PsaQ with other PsbQ(-like) proteins.

2). Although the feature of the four-helix-bundle protein (named as PsaQ here) is similar to PsbQ, this feature is general in PSII including PsbQ, Psb31, Psb27., etc. Thus, the naming of this subunit still needs further consideration.

Response: Thank you for this important suggestion. Here we would like to take the opportunity to explain the reasons of our naming of PsaQ. It is well known that there are 15 common PSI core subunits which were designated PsaA to PsaO. In addition, several other PSI subunits were also discovered. One of these subunits is a transmembrane protein TMP14, which was previously identified as the PSI subunit and designated PsaP (Khrouchtchova, Hansson et al. 2005). The remaining two PSI subunits are only observed in diatom PSI structure, and were designated PsaR and PsaS (Xu, Pi et al. 2020). Furthermore, the structure of PsaQ shows that it possesses the four-helix-bundle architecture similar to PsbQ and PsbQ-like proteins. Therefore, we designated this newly identified four-helix-bundle protein PsaQ in alphabetical order. We have revised our manuscript to explain our naming of PsaQ (Line 95-98).

3. About the RedCAP subunit: The RedCAP is also found in PpPSI-LHC, and the binding location seems to be similar. It would be helpful if the authors could compare the structures of RedCAP subunits between cryptophyte PSI and the red algal PSI-LHC.

Response: Thank you for your suggestion, we have compared the structures of RedCAP subunits between cryptophyte PSI and the red algal PSI-LHC. The result has been shown in Fig. S11 in the revised manuscript.

4. About the PsaQ subunit: As cryptophytes are evolutionarily originated from red algae. Structurally, the PsaQ is similar to PsbQ. Therefore, is the PsaQ genetically derived from some subunits in red algae or cyanobacteria?

Response: Thank you very much for this question, we have also tried for a long time to find an answer but not successful yet. To find the genetic origin of PsaQ, we first blast the protein sequence libraries of red algae and cyanobacteria using PsaQ sequence, but did not find any protein with significant similarity (Response Fig. 6a). In addition, we

used the DALI server to search the PDB for proteins with similar structure of PsAQ. The protein with the highest Z-score (11.4) is PsbQ from cyanobacterium *Synechocystis* sp. PCC6803 (PDB ID 3LS1) (Response Fig. 6b). Its sequence shows approximately 20% similarity with PsAQ sequence (Response Fig. 6c), but it does not bind any pigments. Therefore, without further evidence, we are currently unable to identify the protein in red algae and cyanobacteria as the genetic origin of PsAQ.

Response Fig. 6. The homologue search of PsAQ.

a. The result of blasting PsAQ sequence in protein sequence database of cyanobacteria and red algae. **b.** The result of PsAQ structural homolog searched using DALI server. The structure with the highest score is highlighted by red box. **c.** Sequence alignment between RsPsAQ and PsbQ from the cyanobacterium *Synechocystis* sp. PCC6803.

5. About the hypothetic RsPE545-PSII-PSI-CAC model: The antennae PE545 are densely stacked in the lumen, and PSII and PSI are embedded into the membranes. The authors proposed a hypothetic RsPE545-PSII-PSI-CAC model according to the

dimension of the luminal groove (Fig. 5). However, it is likely that the most of PE545 are located away from the membranes, where the PE545 does not seem to bind to PSII and PSI that on the membranes. Thus, the RsPE545-PSII-PSI-CAC may not stable in native content.

Response: Thank you for your helpful suggestion. We completely agree with the reviewer that the RsPE545-PSII-PSI-CAC may not be stable in native content. We have thus modified our manuscript, clarifying that the RsPE545-PSII-PSI-CAC is only a hypothetic model which requires further exploration as following “*Nevertheless, it is also possible that PE545 may be located distantly from the thylakoid membranes, and the RsPE545-PSII-PSI-CAC is not stable in vivo. To confirm the hypothetic RsPE545-PSII-PSI-CAC model, further experimental evidences are required*”.

6. The PsaQ is associated/disassociated to PSI core under L- and S-phase, and it interacts with CAC-I and PsaB. In addition to providing an efficient energy transfer pathway, will the absence of PsaQ affect changes in the overall structure of PSI core? The authors may be able to make a detailed comparison of related pigments and protein structures in L- and S-phase.

Response: Thank you for your suggestion. Three of our four cryo-EM data sets, namely PSI-14CAC_{L-phase}, PSI-14CAC_{S-phase} and PSI-11CAC_{S-phase}, were collected on the same microscope (300 kV Titan Krios (FEI)) and thus can be aligned. We therefore compared the structures of PSI-14CAC_{L-phase} and PSI-14CAC_{S-phase}, and found no obvious changes in the PSI core regarding both protein subunits and pigment molecules (Response Fig. 7). We hypothesize that the underlying robustness of the PSI core may rationalize this observation.

Response Fig. 7. Superposition of the PSI core between PSI-14CAC_{L-phase} and PSI-14CAC_{S-phase}

structures.

a. Comparison of protein subunits. b. Comparison of pigment molecules. The PSI-14CAC_{L-phase} structure is colored by chains and the PSI-14CAC_{S-phase} structures is in grey.

Reference

- Cheragi, O., E. Kotabova, O. Prasil, W. P. Schroder, R. Kana and C. Funk (2015). "Presence of state transitions in the cryptophyte alga *Guillardia theta*." *J Exp Bot* **66**(20): 6461-6470.
- Engelken, J., H. Brinkmann and I. Adamska (2010). "Taxonomic distribution and origins of the extended LHC (light-harvesting complex) antenna protein superfamily." *BMC Evol Biol* **10**: 233.
- Heidenreich, K. M. and T. L. Richardson (2020). "Photopigment, Absorption, and Growth Responses of Marine Cryptophytes to Varying Spectral Irradiance." *J Phycol* **56**(2): 507-520.
- Kana, R., O. Prasil and C. W. Mullineaux (2009). "Immobility of phycobilins in the thylakoid lumen of a cryptophyte suggests that protein diffusion in the lumen is very restricted." *FEBS Lett* **583**(4): 670-674.
- Khrouchtchova, A., M. Hansson, V. Paakkari, J. P. Vainonen, S. Zhang, P. E. Jensen, H. V. Scheller, A. V. Vener, E. M. Aro and A. Haldrup (2005). "A previously found thylakoid membrane protein of 14kDa (TMP14) is a novel subunit of plant photosystem I and is designated PSI-P." *FEBS Lett* **579**(21): 4808-4812.
- Ludwig, M. and S. P. Gibbs (1989). "Localization of phycoerythrin at the luminal surface of the thylakoid membrane in *Rhodomonas lens*." *J Cell Biol* **108**(3): 875-884.
- Xu, C., X. Pi, Y. Huang, G. Han, X. Chen, X. Qin, G. Huang, S. Zhao, Y. Yang, T. Kuang, W. Wang, S. F. Sui and J. R. Shen (2020). "Structural basis for energy transfer in a huge diatom PSI-FCPI supercomplex." *Nat Commun* **11**(1): 5081.
- Yamamoto, S., P. Bossier and T. Yoshimatsu (2020). "Biochemical characterization of *Rhodomonas* sp. Hf-1 strain (cryptophyte) under nitrogen starvation." *Aquaculture* **516**.

REVIEWERS' COMMENTS:

Reviewer #1 (Remarks to the Author):

The authors have addressed all my concentrated and i support the publication of the manuscript.

Reviewer #3 (Remarks to the Author):

The authors have satisfactorily addressed our comments.

Response to Reviewers' comments:

Reviewer #1 (Remarks to the Author):

The authors have addressed all my concentrated and i support the publication of the manuscript.

Reviewer #3 (Remarks to the Author):

The authors have satisfactorily addressed our comments.

Response: We thank all the reviewers for their valuable suggestions and positive comments.